# The origin of biological homochirality along with the origin of life

**Yong Chen, Wentao Ma**  *

Hubei Key Laboratory of Cell Homeostasis, College of Life Sciences, Wuhan University, Wuhan, China

* mwt@whu.edu.cn

## Abstract

How homochirality concerning biopolymers (DNA/RNA/proteins) could have originally occurred (i.e., arisen from a non-life chemical world, which tended to be chirality-symmetric) is a long-standing scientific puzzle. For many years, people have focused on exploring plausible physic-chemical mechanisms that may have led to prebiotic environments biased to one chiral type of monomers (e.g., D-nucleotides against L-nucleotides; L-amino-acids against D-amino-acids)–which should have then assembled into corresponding polymers with homochirality, but as yet have achieved no convincing advance. Here we show, by computer simulation–with a model based on the RNA world scenario, that the biased-chirality may have been established at polymer level instead, just deriving from a racemic mixture of monomers (i.e., equally with the two chiral types). In other words, the results suggest that the homochirality may have originated along with the advent of biopolymers during the origin of life, rather than somehow at the level of monomers before the origin of life.

## Author summary

People have long been curious about the fact that central molecules in the living world (biopolymers), i.e., nucleic acids and proteins, are asymmetric in chirality (handedness), but as the relevant background, the chemical world is symmetric in chirality. Now that life should have originated from a prebiotic non-life background, how could this dissymmetry have occurred? Previous studies in this area focused their efforts on how the chirality-symmetry may have been broken at the monomer level (i.e., nucleotides or amino acids), but have achieved little advance over decades of years. Here we demonstrate, by *in silico* simulation, that instead, the required chirality-deviation may have been established along with the emergence of biopolymers at the beginning stage in the origin of life–just deriving from a chirality-symmetric monomer pool. The process is actually not only an issue of chemistry but also an issue involving evolution–thus previously difficult to reveal by pure lab work in this area. By modelling the evolutionary process, the present computer simulation study provides significant clues for experiments in future.

**Data Availability Statement:** All relevant data are within the manuscript and its Supporting Information files. Source codes of the simulation program can be obtained from https://github.com/mwt2001gh/The-origin-of-biological-homochirality-along-with-the-origin-of-life/blob/

master/Fig3_nsr_n4.cpp. The version corresponds to the case shown in Fig 4A.

**Funding:** This study was supported by the National Natural Science Foundation of China (No. 31571367) (http://www.nsfc.gov.cn) and Natural Science Foundation of Hubei Province (CN) (No. 2019CFB685) (http://kjt.hubei.gov.cn) to WM. The funders had no role in study design, data collection and analysis, decision to publish, or preparation of the manuscript.

## Introduction

Homochirality is significant for life. Genetic polymers (DNA/RNA) must be composed of residues with the same chirality (handedness) to be able to act as template in replication; functional polymers (proteins/RNA) must be composed of residues with the same chirality to be able to fold into appropriate structures. However, in a prebiotic chemical world, the small molecules from which these macromolecules could be synthesized tend to have existed as racemic mixtures (that is, with equal quantities of the two chiral types). This brings up a significant sub-problem, i.e., the origin of homochirality, for the problem of the origin of life [1–3].

The origin of life is a field full of controversies, which is not surprising when considering we have not even reached a consensus on the definition of life [4–6]. In this field, relevant issues are often not clearly defined. However, the origin of homochirality is an exception–as mentioned above, it is 'straightforward and inevitable', such that apparently more attention has been paid to this issue than to other ones concerning the origin of life. Just because people realized that homochirality is crucial to biopolymers, almost all previous studies in this area assumed that homochirality originated before the emergence of biopolymers [2,3]. That is, the focus has been placed upon investigating plausible physic-chemical mechanisms that may have led to prebiotic environments asymmetric in chirality. It was believed such asymmetry, if initially not in regard of, should have ultimately brought about chirality-bias on 'biomonomers' (i.e., nucleotides and amino acids), which would then assemble into corresponding biopolymers with uniform chirality. Unfortunately, many years of efforts along this line have not led to any convincing conclusion–actually, there are quite a lot of hypotheses or speculations; many of those experimental results are yet far from being relevant to the biomonomers (see [3,7,8] for recent reviews).

Surely, considering that homochirality is so important for the genetic and functional roles of biopolymers, the feasibility that homochirality arose after the emergence of the biopolymers should be discarded. However, actually, the possibility of 'homochirality arising together with biopolymers' cannot be ruled out (see [9] for a discussion). Obviously, such concomitant emergence, if ever occurred, should have been an evolutionary process–with ongoing transitions over a period of time, which is difficult to investigate by pure lab work. That is the chance for *in silico* simulation studies, which are apt at exploring the evolutionary dynamics involved in the origin of life [10–12]. Indeed, here we show by computer simulation–with a model based on the RNA world scenario–that the biased-chirality could have been established at polymer level, just deriving from a racemic mixture at the level of monomers.

The RNA world has been widely accepted as a likely scenario at the beginning of life [13–16]. In this world, the formation of RNA with uniform handedness was critical because RNA acted as both genetic molecules and functional molecules; but its building blocks, nucleotides (and their precursors) should have existed as a racemic mixture then. As the present computer simulation study shows, the tendency of RNA to incorporate monomers with the same handedness as its own (known as 'chiral selection') [17] in polymerization (including the template-directed synthesis and the surface-mediated synthesis; see below for explanation) endowing the system with an autocatalytic property, which may amplify the slight difference between the two chiral forms–perhaps initially occurring by chance. As a result, RNA chains with uniform handedness residues may appear, rendering the appearance of ribozymes possible. The advent of relevant ribozymes, in turn, could enhance the chirality-deviation–which would bring about longer RNA chains with uniform handedness and thus chances for the emergence of more complex and efficient ribozymes. See Fig 1 for a schematic illustration of these ideas. See Methods for an explanation of the model system.

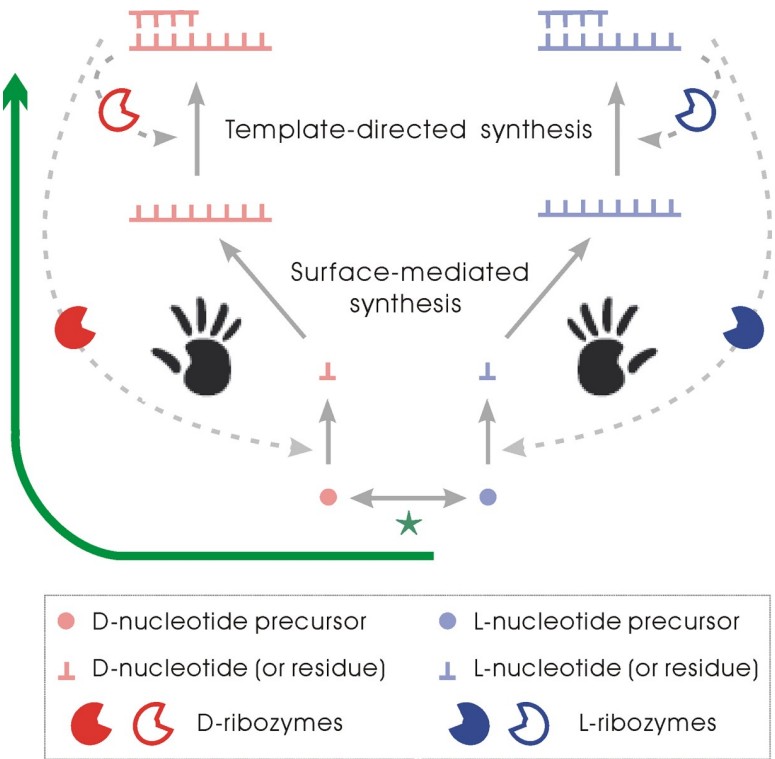

**Fig 1. The arising of homochirality at the polymer level in the RNA world.** The evolution starts with a racemic pool of nucleotide precursors, in which the two chiral types can interconvert readily (the green star). RNA's preference to incorporate monomers of chirality identical to its own ('chiral selection') in its *de novo* polymerization (the surface-mediated synthesis) and replication (the template-directed synthesis) brings about the autocatalytic feature, which may amplify the slight difference between the two chiral forms–perhaps initially occurring by chance. During the amplification, the materials in the opposite form shift toward the target form through the racemization-balancing (the green star). Consequently, substantial chirality-deviation of the whole system would occur (the green arrow), engendering long RNA chains with uniform handedness, which renders the appearance of ribozymes possible. Then, the advent of ribozymes (see text for explanations of such ribozymes), may further enhance the chirality-deviation, due to their more specific, efficient chiral-selection. That is, the resulted biological chirality of one type, instead of the other type, should have initially occurred by chance this way.

## Results

There have long been suspicions that the template-directed synthesis, an obvious autocatalytic process, may have driven the chirality-symmetry breaking in the RNA world [17]. So we investigate this mechanism first. It is assumed that only the monomers with the same chirality as the RNA template can be incorporated into the extending complementary chain. The simulation is initialized with equal quantities of L and D-nucleotide precursors (both half of $T_{NPB}$, see *Table 1* for descriptions of parameters). Some D-RNA (i.e., RNA with D-nucleotide residues) molecules are inoculated into the system a period of time later. Then the symmetry-breaking occurs (Fig 2A, solid line), and the D-enantiomers (summing up over all nucleotide precursors, nucleotides and nucleotide residues in RNA) take up the majority of total-materials. As for the reason behind the process, template-directed synthesis started when the D-templates were inoculated, which consumed D-nucleotides rapidly. Due to the equilibrium between D-nucleotides and D-nucleotide precursors (associated with $P_{NF}$ and $P_{ND}$), D-nucleotide precursors decreased correspondingly. Then, L-nucleotide precursors decreased because of the racemization-balancing process between D-nucleotide precursors and L-nucleotide precursors (associated with $P_{CIC}$; refer to the green star in Fig 1). As a net outcome, there is a large

**Table 1. Parameters used in the computer simulation.**

| Probabilities | Descriptions | Default Values * |
|---|---|---|
| $P_{AT}$ | An RNA template attracting a substrate (nucleotide or oligomer) | 0.5 |
| $P_{BB}$ | A phosphodiester bond breaking within an RNA chain | $1\times10^{-5}$ |
| $P_{CIC}$ | The chirality inter-conversion of nucleotide precursors | 0.5 |
| $P_{FP}$ | The false base-pairing when an RNA template attracts a substrate | 0.001 |
| $P_{MN}$ | The movement of nucleotides | $1\times10^{-4}$ |
| $P_{MPN}$ | The movement of nucleotide precursors | 0.002 |
| $P_{ND}$ | A nucleotide decaying into its precursor | 0.01 |
| $P_{NDE}$ | A nucleotide residue decaying at RNA's chain end | $1\times10^{-4}$ |
| $P_{NF}$ | A nucleotide forming from its precursor (non-enzymatic) | 0.001 |
| $P_{NFR}$ | A nucleotide forming from its precursor catalyzed by NSR | 0.2 |
| $P_{RL}$ | The Random ligation of nucleotides and RNAs (surface-mediated) | $2\times10^{-6}$ |
| $P_{SP}$ | The separation of a base pair | 0.3 |
| $P_{TL}$ | The template-directed ligation (non-enzymatic) | 0.002 |
| $P_{TLR}$ | The template-directed ligation catalyzed by REP | 0.9 |
| **Others** | **Descriptions** | **Default Values *** |
| $N$ | The system is defined as an N × N grid | 20 |
| $T_{NPB}$ | Total nucleotide precursors introduced in the beginning | 50000 |
| $C_T$ | Collision times of nucleotides and RNAs within a grid room | 8 |
| $F_{CSS}$ | The factor for chiral selection in surface-mediated synthesis | 0.5 |
| $F_{CST}$ | The factor for chiral selection in template-directed synthesis | 0.5 |
| $CS_{REP}$ | The characteristic (catalytic domain) sequence of REP | GUUCAG |
| $CS_{NSR}$ | The characteristic (catalytic domain) sequence of NSR | ACUGGC |

* Generally, the simulation cases shown here adopt the default parameter-values, unless being explicitly stated to be different. But note that $P_{NFR}$, the rate associated with the function of NSR (nucleotide synthetase ribozyme), adopts the corresponding default value only when modeling the behavior of NSR, whereas in other cases, it is set to 0; similarly, $P_{TLR}$, the rate associated with the function of REP (RNA replicase ribozyme), adopts the corresponding default value only when considering the behavior of REP, whereas in other cases, it is set to 0. See text for a detailed explanation of these ribozymes.

excess of D-type over L-type at the polymer level, accounting for the overall enantiomeric excess ('ee' for short; this is a widely-used term in this area, meaning one handedness exceeding the other) (Fig 2A, solid line). This process is autocatalytic in respect of chirality because newly-generated D-RNA will act as template and favor further formation of RNA of the same chiral type. However, the ee cannot extend to 1 or -1, which means 'pure chirality', because there are also degradation events. For instance, when D-nucleotide precursors form from the degradation of D-RNA, they may transform into L-nucleotide precursors. Indeed, when the rates related to the RNA's degradation are adjusted upwards, the resulting ee becomes lower; whereas the chirality-deviation is strengthened when the rates associated with the efficiency of template-directed synthesis are tuned up (S1 Fig).

Notably, Higgs' group highlighted the plausibility of the origin of homochirality along with the origin of life previously [9]. In particular, with a quite different model (more abstract than ours here), they also showed that the chiral selection in template-directed synthesis of RNA may lead to chirality-deviation [18], which demonstrates the robustness about the conclusion on the role of this autocatalytic process in inducing the chirality-deviation. A key puzzle is about the 'cross-inhibition' effect–in that famous experiment [17], while demonstrating the chiral selection for RNA's template-direct synthesis, it was shown that the selection cannot be

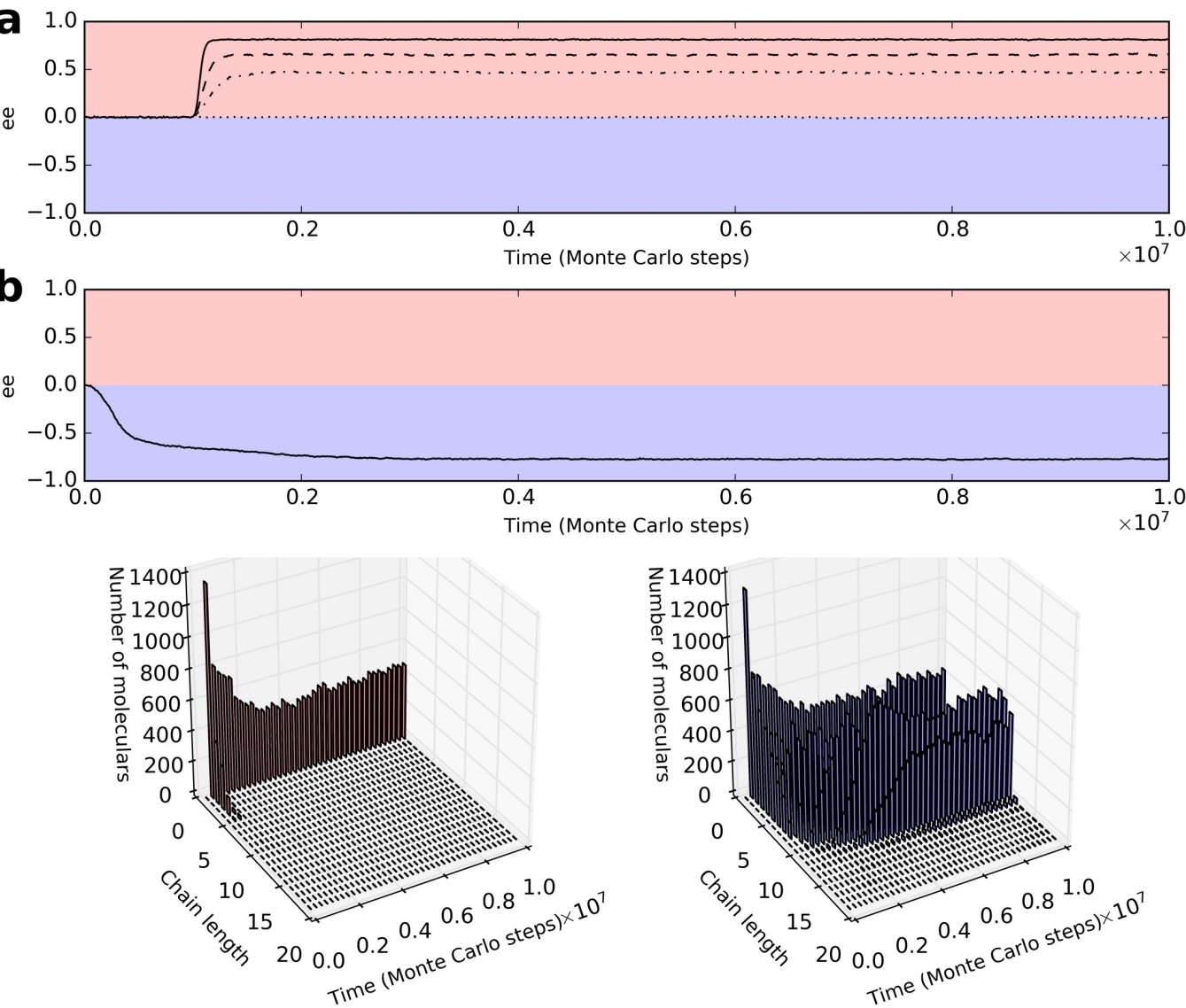

**Fig 2. Chirality-deviation may result from the template-directed synthesis.** Color legends: Red for D-type and blue for L-type (applicable in all figures of the paper). Enantiomeric excess ('ee') equals to $(D-L)/(D+L)$, where $D$ and $L$ represent corresponding enantiomers summed up over all the nucleotide precursors, nucleotides and nucleotide residues in RNA within the system. $P_{TL} = 0.01$. (**a**) 50 molecules of D-RNA, 6 nt in length, are inoculated at $1\times10^6$ step. $P_{RL} = 0$ (i.e., *de novo* appearance of RNA is impossible). Solid line: $F_{CST} = 0$ (complete chiral-selection); dashed line: $F_{CST} = 0.5$ (partial chiral-selection); Dash-dotted line: $F_{CST} = 1$ (no chiral-selection). See S2 Fig for the cases corresponding to more $F_{CST}$ values. The dotted line represents the case in which neither chiral-selection nor cross-inhibition termination exists. (**b**) RNAs appear *de novo*. $P_{RL} = 1\times10^{-6}$, $F_{CST} = 0.5$, $F_{CSS} = 0$. This is a case in which L-RNA prevails–which is in practice 'by chance' (e.g., if using a different random seed in the Monte-Carlo simulation, D-RNA might prevail). The evolution regarding RNA's chain-length distribution is displayed below, respectively of the two chirality types.

a complete one, which means nucleotides with the opposite handedness also have a chance to incorporate; moreover, if an 'opposite' monomer joins the extending complementary chain, the template-directed synthesis will terminate. In fact, this effect not only throws doubts on the possibility of the origin of homochirality in this way, but also shakes the cornerstone of the RNA world scenario itself because RNA's replication is blocked.

So then, we take the incompleteness of chiral-selection and the cross-inhibition into account. As expected, a lower rate of the chiral selection (i.e., a larger $F_{CST}$, see Methods) results in a lower ee (S2 Fig). But somewhat unexpectedly, the autocatalytic process seems to

be rather robust against the cross inhibition. For instance, even when $F_{CST} = 0.5$, which means a monomer with an opposite handedness has a chance of 50% to be incorporated, thus stopping the ongoing synthesis, the chirality-deviation may occur as well–with a substantial ee (Fig 2A, dashed line). What surprises us most, however, is that even when the $F_{CST}$ is set to 1, i.e., the template does not distinguish the handedness of substrate nucleotides, the symmetry-breaking still occurs (Fig 2A, dash-dotted line). We speculated that the reason is associated with the termination effect of the cross inhibition. That is, the best situation is certainly that no opposite-handedness monomer would be incorporated, but when the incorporation of opposite-handedness monomer is inevitable, stopping further extension of the chain is the 'best choice'. Indeed, if the termination effect is removed, the chirality-deviation no longer occur (Fig 2A, dotted line)–therein, mosaic chains with a combination of D and L-nucleotides are produced, and the two enantiomers act completely equivalently in the template-directed synthesis, thus resulting in no symmetry-breaking. This result is impressive, because it means that the 'cross inhibition', originally deemed as a negative factor, is, from a certain angle, a positive factor for the arising of homochirality.

Another question is: how could the handedness-uniform templates have emerged *de novo* (note: in the modeling cases above they were inoculated *ad hoc*)? It has long been suggested that in the RNA world, the *de novo* RNA-synthesis was mediated by mineral surface (e.g., the montmorillonite clay) [19–22]. Notably, because relevant minerals could be encapsulated into lipid vesicles, such surface-mediated synthesis of RNA may even have been able to occur in lipid membrane systems [23]–emphasizing the role of surface-mediation in the de novo synthesis of RNA even if the RNA world might have started in a form represented by RNA-based protocells. Relevant to the current topic, interestingly, it was found that in such surface-mediated synthesis, the foregoing residues in an extending RNA chain would favor the incorporation of the same chiral type of monomer as their own [24–26]. Obviously, this is also a sort of chiral selection, and it may have generated initial RNA templates with uniform handedness. Indeed, when such a kind of *de novo* polymerization is allowed, we see a naturally-occurred chirality-deviation–randomly biased to D- or L-type (Fig 2B represents a case in which L-type prevails). In the *de novo* polymerization, RNA of one chiral type may be 'a little more ahead' by chance, and the autocatalytic template-directed synthesis would amplify the distinction. Note that the evolution of RNA's chain-length distribution (Fig 2B, bottom panels) may reflect well 'the establishment of chirality-deviation at the polymer level'.

Like the template-mediated synthesis, in reality, the chiral selection involved in the surface-mediated synthesis of RNA cannot be a complete one, and it was even implied that there is also 'cross-inhibition' during this process [24–26] (note: when we consider the incomplete chiral-selection and cross-inhibition in the surface-mediated synthesis, spontaneous chirality-deviation like the case shown in Fig 2B can still occur in our modeling). In mechanism, for a surface-mediated reaction, it is the residue at the end of a polymer, with the 'structural' aid of mineral surface, that favors the incorporation of a monomer with the same handedness as itself–whereas, for the template-directed reaction, both the template and the residue at the end of the polymer may work in the chiral selection. No matter how, from the principle of chemistry, the surface-mediated synthesis and the template-directed synthesis have 'built-in' similarities. We are then interested in: is the surface-mediated synthesis also able to derive the chirality-deviation, by itself?

Somewhat unexpectedly, along this line, initially, we failed in finding any case of chirality-deviation caused by the surface-mediated mechanism, though we have managed to favor such a positive result (e.g., we have ever tuned the condensation rate, i.e., $P_{RL}$, to a rather high level). Then, we reasoned that the key 'autocatalytic feature' associated with the chiral selection is here in practice invalidated for the surface-mediated synthesis. Actually, what makes the

autocatalytic feature in the template-directed synthesis valid is the superiority of the template-directed synthesis to the surface-mediated synthesis. For instance, to 'catch up with' the 'leading' chiral type (as majority), the 'lagging' chiral type, with a less quantity of templates, has to generate more RNA templates *de novo*, which is far less efficient than the template-directed synthesis. However, here, for the surface-mediated synthesis itself, because the starting and the extending of chain-synthesis are of the same rate, there would be no difference for which chiral type to get ahead.

Then, interestingly, at this point, we noticed that in reality, the extending is actually more efficient than the starting during the polymerization of RNA, as demonstrated experimentally for both the template-directed synthesis [27,28] and the surface-mediated synthesis [29,30], known as the 'primer effect'. Indeed, when considering this mechanism, and if only the primer effect is sufficient strong, the surface-mediated synthesis may also induce the chirality-deviation in itself (Fig 3) and the chirality-deviation is proportional to the rate of relevant chiral selection (S3 Fig). (Note: the primer effect is not so critical for the template-directed synthesis to induce chirality-deviation, as already implied in the reasoning above; practically in our modeling, when considering this effect for template-directed synthesis, no qualitative difference was observed for the cases shown in Fig 2; however, to keep consistent, we consider this

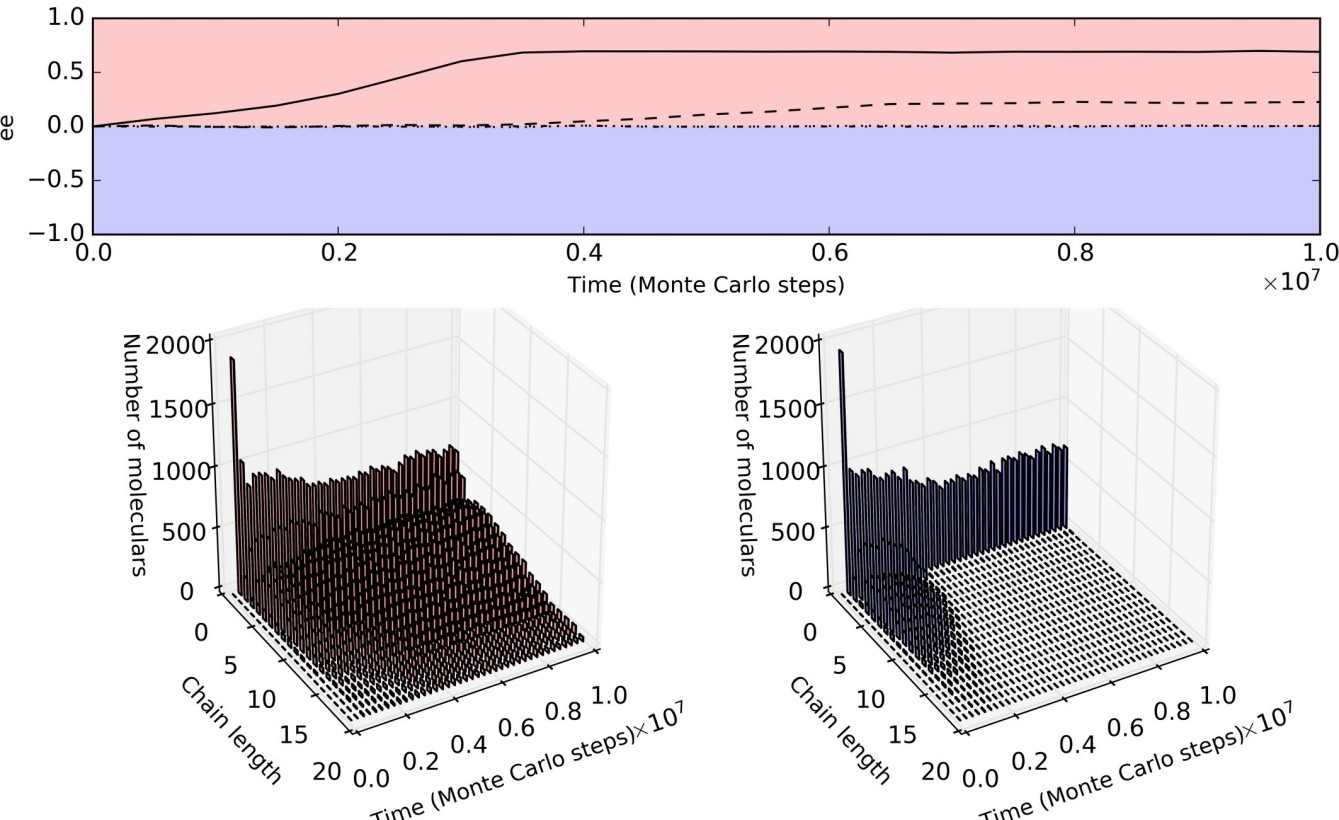

**Fig 3. Chirality-deviation may result from surface-mediated synthesis when considering the primer effect.** $P_{AT} = 0$ (i.e., template-directed synthesis is disabled); $P_{RL} = 1\times10^{-6}$, $P_{NDE} = 1\times10^{-5}$, $F_{CSS} = 0$. The extending rate for a monomer, dimmer, trimmer and longer polymer is represented as $R_{1\text{-mer}}$, $R_{2\text{-mer}}$, $R_{3\text{-mer}}$, and $R_{n\text{-mer}}$, respectively ($P_{RL}$ is multiplied by one of such rates in the corresponding situation). Dotted line (control): $R_{n\text{-mer}} = R_{3\text{-mer}} = R_{2\text{-mer}} = R_{1\text{-mer}}$ (the dotted line actually overlaps with the dashed-dotted line, which are both at the level of ee = 0); dash-dotted line: $R_{n\text{-mer}} = R_{3\text{-mer}} = 5\times R_{2\text{-mer}} = 50\times R_{1\text{-mer}}$ (the primer effect is not sufficiently strong to induce the chirality-deviation); dashed line: $R_{n\text{-mer}} = R_{3\text{-mer}} = 10\times R_{2\text{-mer}} = 100\times R_{1\text{-mer}}$ (the primer effect is strong enough to induce a chirality-deviation); solid line: $R_{n\text{-mer}} = R_{3\text{-mer}} = 20\times R_{2\text{-mer}} = 200\times R_{1\text{-mer}}$ (the primer effect is so strong as to induce a significant chirality-deviation). The evolution of RNA's chain-length distributions for the solid-line case is displayed below (the chains longer than 20 nt are not displayed). Note that the prevailing of D-RNA (instead of L-RNA) as shown in the solid-line case and the dashed-line case is merely by chance.

effect for both the surface-mediated and template-directed syntheses in the modeling hereinafter). In fact, the 'primer effect' means that polymers already formed would favor further polymerization (like the situation in the template-directed synthesis), thus endowing the surface-mediated synthesis with the autocatalytic feature which is important for the breaking of the chirality-symmetry.

Accordingly, the template-directed synthesis and the surface-mediated synthesis can trigger the symmetry-breaking independently. Here, a significant point is that by demonstrating the two results in parallel, we see that the establishment of chirality-deviation at polymer level is well plausible. In reality, it is possible that the two mechanisms are cooperative, thus making the chirality-deviation more robust, rendering the emergence of long RNA with uniform handedness possible, which is significant for the subsequent emergence of functional ribozymes.

An RNA species catalyzing the template-directed synthesis (thus the RNA replication) has been long suggested to have been the first ribozyme emerging in the RNA world, usually called an 'RNA replicase' [31–33] (here 'REP' for short). This plausibility has been supported by *in silico* simulation [34,35]. Alternatively, a ribozyme capable of catalyzing the synthesis of RNA's building block [36], i.e., a nucleotide synthetase ribozyme ('NSR'), can also favor its own replication by supplying the monomers and thus may (in principle of Darwinian evolution) as well have spread in the RNA pool initially–the plausibility of which has also been supported by computer modeling [37]. Here we are curious about: whether could these ribozymes, given the request of chirality-uniformity for their chains, emerge in a chirality-deviation background established by the surface-mediated synthesis and the template-directed synthesis?

NSR is here assumed to catalyze the synthesis of nucleotides with the same chirality as its own. It turns out that this ribozyme may emerge spontaneously in the model system, and furthermore, ee increases significantly with its advent (Fig 4A, Fig 5 and S1 Movie). In fact, NSR consumed nucleotide precursors of its own chirality-type, thus inducing further chirality-deviation via the racemization-balancing process of nucleotide precursors (see the green star in Fig 1). REP is here assumed to catalyze the template-directed synthesis of RNA with chirality-type identical to the ribozyme itself. It turns out that this ribozyme may also emerge spontaneously, and then the ee increases apparently (Fig 4B). This outcome is also reasonable: as we have shown, the enhanced efficiency of the template-directed synthesis would lead to the corresponding enhancement of the chirality-deviation (S1 Fig). Interestingly, there was recent experiment work showing that a REP may catalyze the template-directed synthesis of the opposite-chiral-type RNA [38]. This is understandable according to our chemical knowledge, and in principle there should also be NSR capable of catalyzing the synthesis of nucleotides with the opposite chirality to its own. However, these 'opposite ribozymes' could not emerge in the system when considering the chirality context, because they would disfavor their own replication given the competition on raw materials (nucleotide precursors) between the two chiral types (via the racemization-balancing, see the green star in Fig 1). Finally, we noticed that with the chirality-deviation enhancement (induced by the ribozymes), the RNA's chain-length with uniform handedness increases (Fig 4A and 4B, the chain-length-distribution panels), which would provide opportunities for the subsequent emergence of more complex and efficient ribozymes. This may represent a sign for the ongoing inter-promotion between the developments of the living world and the relevant chirality-deviation.

## Discussion

According to the conceptual analysis from Higgs' group [9], the arising of homochirality at the same time as the origin of life means that 'there is no asymmetry in the monomers' and

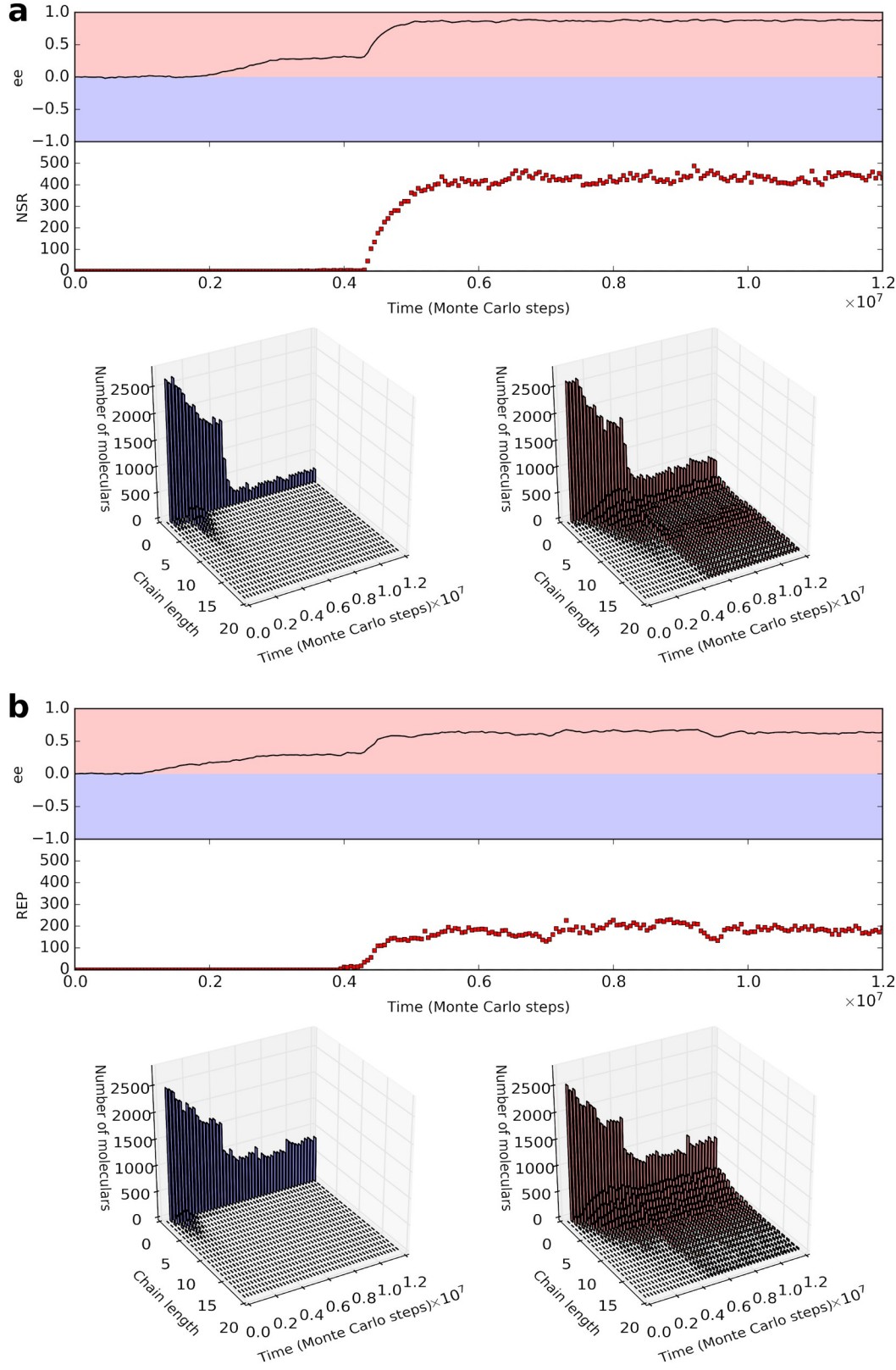

**Fig 4. The emergence of ribozymes in the chirality-deviation background and subsequent enhancement of the chirality-deviation.** (**a**) The spontaneous appearance of NSR (with a uniform-handedness catalytic domain) in a chirality-

deviation background established by the surface-mediated synthesis and the template-directed synthesis, and its effect of augmenting the enantiomeric excess. REP is not considered (i.e., $P_{TLR} = 0$). About the primer effect in surface-mediated synthesis: $R_{n\text{-mer}} = R_{3\text{-mer}} = 2 \times R_{2\text{-mer}} = 20 \times R_{1\text{-mer}}$ ($P_{RL}$ is multiplied by one of such rates in corresponding situation; the scale of these rates are generally in accordance with experiments [29,30]; about the primer effect in template-directed synthesis: $R_{n\text{-mer}} = R_{3\text{-mer}} = 2 \times R_{2\text{-mer}} = 10 \times R_{1\text{-mer}}$ ($P_{TL}$ is multiplied by one of such rates in corresponding situation; the scale of these rates are generally in accordance with experiments [27,28]). See Fig 5 and S1 Movie for the evolutionary scenario of this case. (**b**) The spontaneous appearance of REP (with a uniform-handedness catalytic domain) in a chirality-deviation background established by the surface-mediated synthesis and the template-directed synthesis, and its effect of augmenting the enantiomeric excess. NSR is not considered (i.e., $P_{NFR} = 0$). The situation about the primer effects are the same as that assumed in (*a*). For the panels of chain-length distribution, the chains longer than 20 nt are not displayed. Note that the prevailing of D-RNA and corresponding ribozymes shown in these two cases is merely by chance (i.e., in other cases L-type ones might show up).

'autocatalytic biopolymers create a chiral bias'. The scenario described here is just such a kind of bio-homochirality origin, considering the autocatalytic feature of template-directed synthesis and surface-mediated synthesis (under the primer effect), as well as the autocatalytic feature of ribozyme-aided reactions. In later papers of theirs [18,39], it was elucidated that the polymer synthesis and evolution before the emerge of ribozymes should be summarized into the 'chemical evolution' and 'does not quite constitute life' [18,39]. If so, the present paper is suggesting that the chirality-symmetry breaking occurred at polymer level before the origin of life, and promoted the emergence of initiate functional polymers (with homochirality), which then amplified the chiral bias–during the origin of life. However, if we accept that the origin of life just started with the chemical evolution and then proceeded to biological evolution, and in the sense that the arising and subsequent amplification of the chiral bias as described here just covered such an 'interface between life and non-life', it is still quite reasonable to phrase the scenario as 'the origin of biological homochirality along with the origin of life'.

With an idea that homochirality may have originated at monomer level, some researches focused on the formation of pure-chirality crystal from racemic solution of small molecules, via a process named 'chiral amnesia' [3,7]. A major problem regarding this thought is: even if pure-chirality monomers can assemble this way, to take part in polymerizing reaction to give rise to biopolymers, they 'have to' re-dissolve into the solution, wherein racemization is inevitable. While doubting the relevance of this thought to the origin of biological homochirality, we note that the formation of homochirality polymers as described in the present study is in principle very similar to the 'chiral amnesia' process. As we know, the two types of enantiomers are almost identical in all chemical/physical aspects, and it is rather difficult to 'choose' one type but 'ignore' the other by ordinary physic-chemical mechanisms. Chiral crystallization provides a possible solution: that is, let the small molecules (those already in the crystal) choose themselves (others in the solution), and the racemization-balancing in the solution would turn those molecules of the opposite type into the 'right type' [3]. Similarly, during the polymerization of RNA, either template-directed or surface-mediated, it is the monomers (residues) already in the polymer that select the 'right' monomers to be incorporated in the next step, and the racemization of nucleotide precursors in the circumstance would transfer the opposite type into the 'right' one (see Fig 1). Interestingly, similar ideas were expressed in a recent review in this area, proposing the possibility of formation of chirality-deviation at the polymer level (though presented in a more abstract context–without a concrete scenario associated with nucleotides and RNA) [40].

Here, we explore the mechanisms concerning the arising of biological homochirality at polymer level based on the scenario of the RNA world. If life really began with an RNA world and subsequently transformed to a DNA/RNA/proteins world, DNA may have inherited the homochirality of RNA naturally (as D-type) by template-directed synthesis, but the arise of L-proteins (with uniform L-amino-acid residues) may be more complicated–perhaps with the

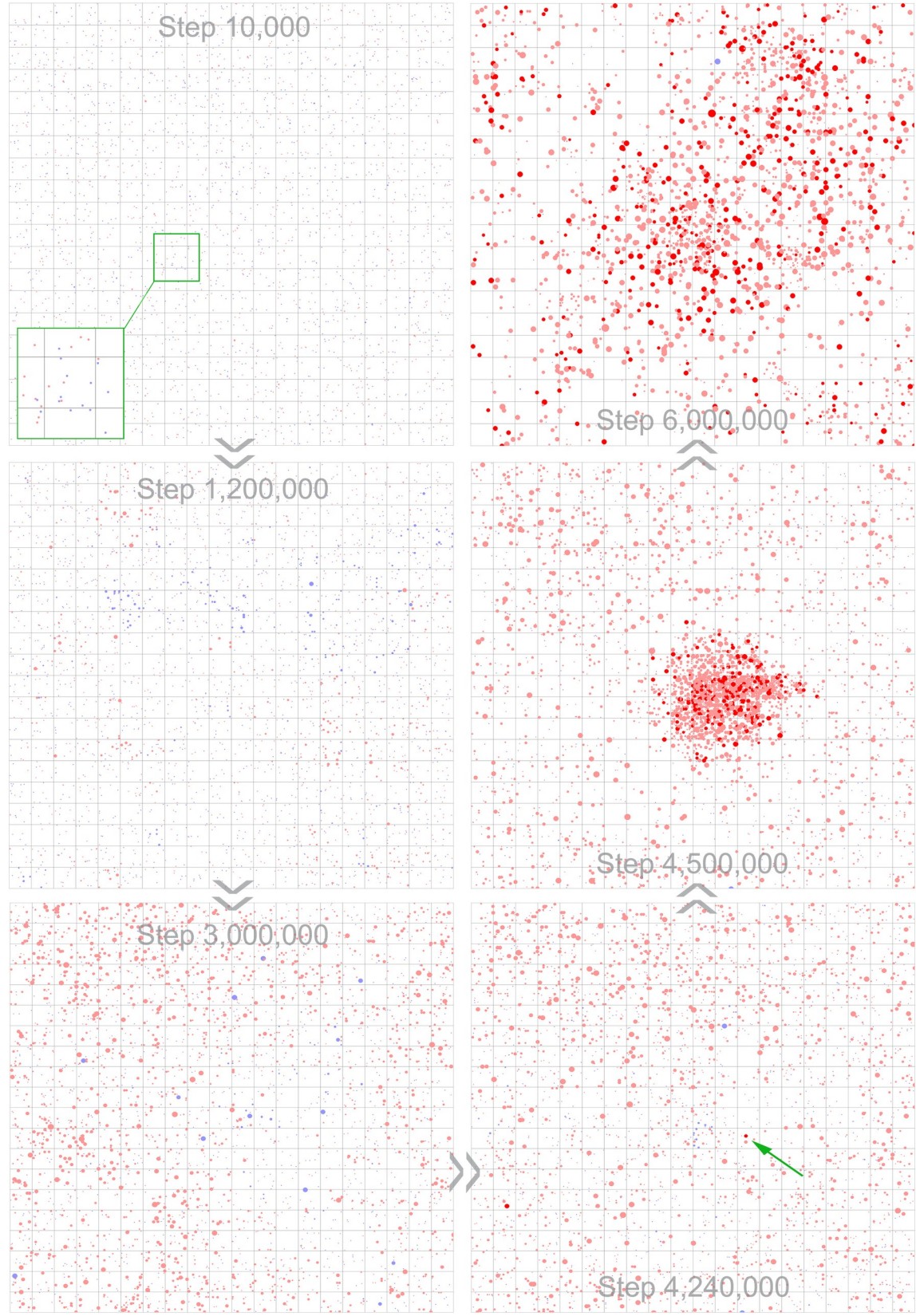

**Fig 5. Snapshots showing the natural arising of chirality-deviation induced by the surface-mediated synthesis and the template-directed synthesis and subsequent spontaneous emergence of NSR.** Molecules of nucleotides and RNA are represented as solid circles (dots), with diameter in proportion to the square root of the chain-length of these molecules. D-type of nucleotides and RNA are denoted in light red, except for D-NSR, which is denoted in bright red. L-type of nucleotides and RNA are denoted in light blue, except for L-NSR, which is denoted in bright blue (but note: in this case no L-NSR emerges). Step 10,000: after inoculation of nucleotide precursors in the beginning (Step 0), many nucleotide molecules of both chiral types form (tiny dots, see the zoom-in panel); Step $1.2 \times 10^6$: the formation of oligomers of both chiral types; Step $3 \times 10^6$: one chiral type (D-type in this case) achieves superiority, resulting from the surface-mediated synthesis and the template-directed synthesis; Step $4.24 \times 10^6$: the NSR molecule (see the green arrow) which ultimately gives rise to the thriving of NSR in the whole system; Step $4.5 \times 10^6$: the spread of the NSR; Step $6 \times 10^6$: the thriving of the NSR in the whole system. What is displayed in S1 Movie is of the same case, which focuses on the appearance and spread of the NSR (from step $4.2 \times 10^6$ to $4.7 \times 10^6$). See Fig 4A for the evolutionary dynamics of the case.

participation of special ribozymes, depending on the process and mechanisms involved in the emergence of 'translation machine'. Alternatively, as we have noted, the logic behind the idea of the RNA world is actually not unshakable (for a detailed discussion, see [16]). If life was in fact initiated with, say, an RNA/proteins world, the emergence of L-proteins may also have involved certain process(es) concerning the arise of homochirality at polymer level–deriving from racemic amino acids or their precursors, perhaps in some aspects similar to the scene described here on RNA (e.g., by surface-mediated synthesis; but note that template-directed synthesis should not be available for proteins). Therein, there may have been interactions between the two chirality-deviation processes–concerning RNA and proteins respectively.

It deserves mention that in history, there have been a series of theoretical studies concerning the origin of bio-homochirality accompanying the appearance of biopolymers (e.g., [9, 41–48]), which largely derived from the seminal work of Sandars (2003) [41]. However, such modeling studies tackled this issue in a rather abstract way. For example, they even did not consider template-directed synthesis of the polymers, which played a key role in both chemical evolution and biological evolution (i.e., Darwinian evolution, based upon molecular replication). In the models, the polymerization runs only in a non-templated way (i.e., the *de novo* synthesis). Another remarkable simplification is that there are not sequence-requirement for the functional polymers, which were assumed to catalyze the synthesis of monomers with the same handedness as their own (thus, with a role somewhat like the NSR in the present study). Sandars assumed that polymers with the maximum length allowed in his model can act as such catalysts [41], and modified model versions commonly assumed that all kind of polymers (regardless of lengths) contributed to the overall catalytic efficiency [42–45,47]. Such assumptions went against general 'biological senses'–perhaps because the polymerization concept here is extended from a classic chemical model [49], which address solely a special problem of dimerization (in which the monomers themselves act as auto-catalysts). Therefore, the modellings should be, at least, doubtful in respect of their relevance to reality in the origin of life. On the other side of coin, however, these abstract models (the 'toy models', as Sandars, the author of the seminal work of this series, called it [41]), do have provided some conclusions shared with the present study. For instance, as mentioned already, the role of those functional polymers is actually similar to that of NSR here–therefore, it is not surprising to see that when spatial limitation is considered, both such modellings [47,48] and ours (Fig 5, step 4,500,000) revealed a localizing pattern (i.e., spatial limitation may favor the formation of chirality-deviation). In mechanism, the functional polymers (with homochirality), either the abstract ones in their studies or the NSR in our study, may catalyze the formation of monomers with the same handedness as themselves, and spatial limitation would prevent the formed monomers from fast diffusing away and thus favor the synthesis of the functional polymers themselves–and therefore, promoting further chirality-deviation.

Notably, a classic experimental work from Eschenmoser's group [50] showed that tetramers of pyranosyl-RNA (an RNA-like polymer) containing 'hemi self-complementary base

sequence' may self-assemble in a chirality-selective way (by 'cross-templating') and ultimately give rise to higher oligomers with uniform chirality. The key point is that the original tetramers actually exists as a mixture of all kinds of diastereomers (i.e., most of the tetramers are D-L chimeras). The results led the authors to suggest that biomolecular homochirality may have, in mechanism, originated this way–it was deduced in principle that when the co-oligomerization exceeds a critical level the chirality-symmetry may break (for a detailed discussion, see the original paper). This study is interesting. At least, as also mentioned by the authors [50], it provides a mechanism to the formation of the initiate homochirality template (in respect of the present modeling, an alternative to the surface-mediated synthesis), which allows the following 'operation' of template-directed synthesis in inducing the chirality-deviation. The key doubt is that the substrates are length-fixed (tetramers) and sequence-dependent (hemi self-complementary), which is quite unrealistic. Though the authors have defended for these requirements in some degree, it is not convincing that the scenario in reality may evolve as they anticipated. That is, an important question is that whether such chirality-selective self-assemble of oligomers could still occur when the requirement of their length-fixed and sequence-dependent features is released. Corresponding modeling studies would be interesting and may direct further experimental work.

The origin of life is a procedure combining chemistry and evolution [51]. Here the evolution is in a general sense, conceptually comprising the so-called 'chemical evolution' and the initial stages of Darwinian evolution. Now computer simulation (modeling) has becomes a significant approach in the field of the origin of life [10–12]–the very reason is just that it is apt at tackling the evolutionary aspect of the procedure. The doubt concerning the anticipation of Eschenmoser's group [50], in regard of the outcome when the requirement of their length-fixed and sequence-dependent features is released, just involves certain complicated evolution process. As the scene suggested in the present study, the arising of homochirality at the polymer level is still more a typical case demonstrating the significance of evolution. Indeed, about 35 years ago, chemists in this area already attempted to assess this scene in laboratory, but the 'cross-inhibition' in the template-directed synthesis of RNA [17] puzzled them and restricted their imagination. In the present modeling study, we show that, given a process of evolution, the 'cross-inhibition' should not have blocked the way along which homochirality could emerge at the polymer level; and it may even be deemed as a positive factor–because otherwise, chirality-chimera polymers would occur and the chiral selection, the driven power for the chirality-deviation, would be largely undermined. Future experimental studies may be conceived with consideration on the results here, e.g., an important notion may be to design some mimic process allowing evolution over a period of time–somewhat like (but perhaps much more complicated than) the one shown in the famous Miller-Urey experiment [52]. If so, that would be a vivid demonstration for how experimental work can be aided by computer simulation to address a problem with the combination of chemistry and evolution–perhaps as an ordinary way of future studies in the field of the origin of life.

## Methods

### The model

We assume a two-dimensional system, with an N × N square grid (with toroidal topology to avoid edge effects), in which molecular objects are distributed, including the raw materials to synthesize nucleotides (namely nucleotide precursors, in quotient of nucleotides), nucleotides, and RNAs. In fact, the two-dimensional model system can just be thought as a mineral surface, which allows the surface-mediated synthesis of RNA (see our previous papers [35,37] for more detailed explanations). Molecules may move to adjacent rooms during the simulation process,

but only molecules within the same 'grid room' are possible to interact with each other within one time step.

Nucleotide precursors with opposite handedness may interconvert. The modeling target of the nucleotide precursors is glyceraldehydes (refer to a significant experiment on prebiotic nucleotide-synthesis [53]), which conduct racemization readily in reality. But note that D-nucleotides and L-nucleotides cannot interconvert directly according our chemical knowledge. A nucleotide precursor may transform to a nucleotide (randomly as A, U, C, or G) with its handedness unchanged, and a nucleotide may also decay into a nucleotide precursor with the same handedness. Nucleotides may join to each other, forming oligomers, and further, longer RNAs–*de novo* polymerization of RNA (i.e., the surface-mediated synthesis). An RNA molecule may attract substrates (nucleotides or oligomers) via base-pairing with certain fidelity (related to $P_{FP}$) and substrates aligned on the template may be ligated (i.e., the template-directed synthesis). Both the surface-mediated synthesis and the template-directed synthesis have the effect of chiral selection (i.e., tending more to incorporate nucleotides with the same handedness) and the effect of cross-inhibition (i.e., the incorporation of nucleotides of the opposite chirality-type would terminate the elongation). Phosphodiester bonds within an RNA chain may break, and thus the RNA molecule turns into two fragments. A nucleotide residue at the end of an RNA chain may decay into a nucleotide precursor with the same handedness. The complementary chain and the template may detach from each other if the base pairs between them separate. Each event occurs with a certain probability in a time step (thus, also referred to as a 'Monte Carlo step'). See *Table 1* for descriptions of the probabilities and Fig 6 for a schematic of the model.

Our modeling approach considers the RNA molecules with a 'resolution' at the level of monomer residues (i.e., discriminating among the four types of bases). This provides convenience for us to conduct relevant studies on the emergence of ribozymes, because we can link (assume) a function to a sequence straightforwardly–as we know, the sequence-function linkage is just the key requirement of Darwinian evolution (which in reality is implemented by the folding of the functional molecules). In practice, here, an RNA containing a characteristic catalytic domain (the sequence being presumed arbitrarily) with uniform chirality is assumed to be able to function as a ribozyme. The nucleotide synthetase ribozyme (NSR) and the RNA replicase (REP) have different catalytic domain sequences. NSR may catalyze the synthesis of nucleotides with chiral type identical to its own (certainly, the raw materials are nucleotides precursors with the same corresponding handedness). REP may catalyze the template-directed ligation of substrates aligned on the template (i.e., the REP is actually a ligase-type one, see [35] for details), provided that the template and the substrates are of the same chiral type as the REP. In the modeling, the catalytic domain of the ribozyme is only 6 nt long. Though there was lab work showing the plausibility of so tiny ribozymes (e.g., see [54]), this seems not likely to be the case for a REP or NSR in reality. Here, we adopted such a short length just to avoid cumbersome computation in the simulation, especially considering the difficulty with regard to demonstrating the *de novo* emergence of these ribozymes (Fig 4). On a related note, the total materials in the system ($T_{NPB}$) here are adopted to be obviously less than those in reality, also in order to limit the computational intension.

## The setting of parameters and some relevant mechanisms in details

The probabilities concerning the events in the system should be set according some rules. Reactions catalyzed by ribozymes should be much more efficient than corresponding non-enzymatic reactions, so $P_{TLR}>>P_{TL}$ and $P_{NFR}>>P_{NF}$. Template-directed ligation should be apparently more efficient than surface-mediated ligation, so $P_{TL}>>P_{RL}$. Here, nucleotide

residues within the chain are assumed to be unable to decay–they should be protected therein, whereas those at the end of the chain, which is 'semi-protected', decay at a rate lower than that of free nucleotides, i.e., $P_{NDE} < P_{ND}$. Considering the efficient racemization of nucleotide precursors (glyceraldehydes), $P_{CIC}$ should be quite high. Other considerations may include: $P_{BB}$ may be higher than $P_{RL}$, but lower than $P_{NDE}$; $P_{MN} < P_{MPN}$; $P_{NF} < P_{ND}$, etc.

With the breaking of phosphodiester bonds, an RNA molecule may degrade into shorter ones (including nucleotides). When the breaking site of the chain is at a single-chain region, the breaking rate is $P_{BB}$. When the breaking site is within a double-chain region, the two parallel bonds may break simultaneously, with the probability $P_{BB}^{3/2}$. The adoption of the index 3/2, instead of 2, corresponds to the consideration of the synergistic effect of the two breaking events.

As mentioned already, nucleotide residues at the end of an RNA chain (either at 3'- or 5'-end) may decay. With similar consideration to RNA's chain-breaking above, when the chain end is in a single-chain status, the decaying rate is $P_{NDE}$; when the chain end is in a double-strand status, the two paired nucleotide residues may decay simultaneously, with a probability of $P_{NDE}^{3/2}$, also in consideration of a synergistic effect.

The probability of the separation of the two strands of a duplex RNA is actually assumed to be $P_{SP}^{(r+1)/2}$, where $r$ is the number of base pairs in the duplex. When $r = 1$, the probability would be $P_{SP}$. When $r$ increases, the probability would decrease (because $P_{SP}$, as a probability, has a value between 0 and 1). That is, the separation of the two strands would be more difficult if the base pairs are more. The introduction of the 1/2 corresponds to the consideration that self-folding of single chains may aid the dissociation of the duplex.

The probability of the movement of an RNA molecule is assumed to be $P_{MN}/m^{1/2}$, where $m$ is the mass of the RNA, relative to a nucleotide. This assumption represents the consideration of the effect of the molecular size on the molecular movement. The square root was adopted here according to the Zimm model, concerning the diffusion coefficient of polymer molecules in solution [55].

In the surface-mediated synthesis, nucleotides with the same handedness as the foregoing residues are incorporated with a rate of $P_{RL}$, whereas nucleotides with the opposite handedness are incorporated with a rate of $P_{RL} \times F_{CSS}$. In the template-directed synthesis, nucleotides with the same handedness as the foregoing residues (surely, also identical to the template's handedness–otherwise, no incorporation would occur because of the cross-inhibition effect) are incorporated with a rate of $P_{TL}$, whereas nucleotides with the opposite handedness are incorporated with a rate of $P_{TL} \times F_{CST}$.

## Other assumptions when implementing the model

In a time step, for a grid room, collision is checked for a certain number of times ($C_T$). Each time, RNA molecules (including nucleotides) in the grid room are grouped pairwise, simulating collision events. Upon a collision, one of the two events may occur: the first, they may ligate end to end with each other; the second, if a molecule (as a substrate) is not longer than the remaining single-chain-length of the other RNA molecule (as the template), it may be attracted onto the latter, aligned next to the last foregoing-substrate–based on the rule of base-pairing (if there is not a foregoing-substrate, this substrate will become the first substrate for the template-directed synthesis).

For the functioning of NSR, each single-chain RNA molecule is checked for the ribozyme's characteristic domain (surely, the subsequence must be in uniform handedness). The molecule is deemed to be an active NSR molecule if it contains the domain but does not exceed double length of the domain (considering that too many redundant residues may seriously interfere

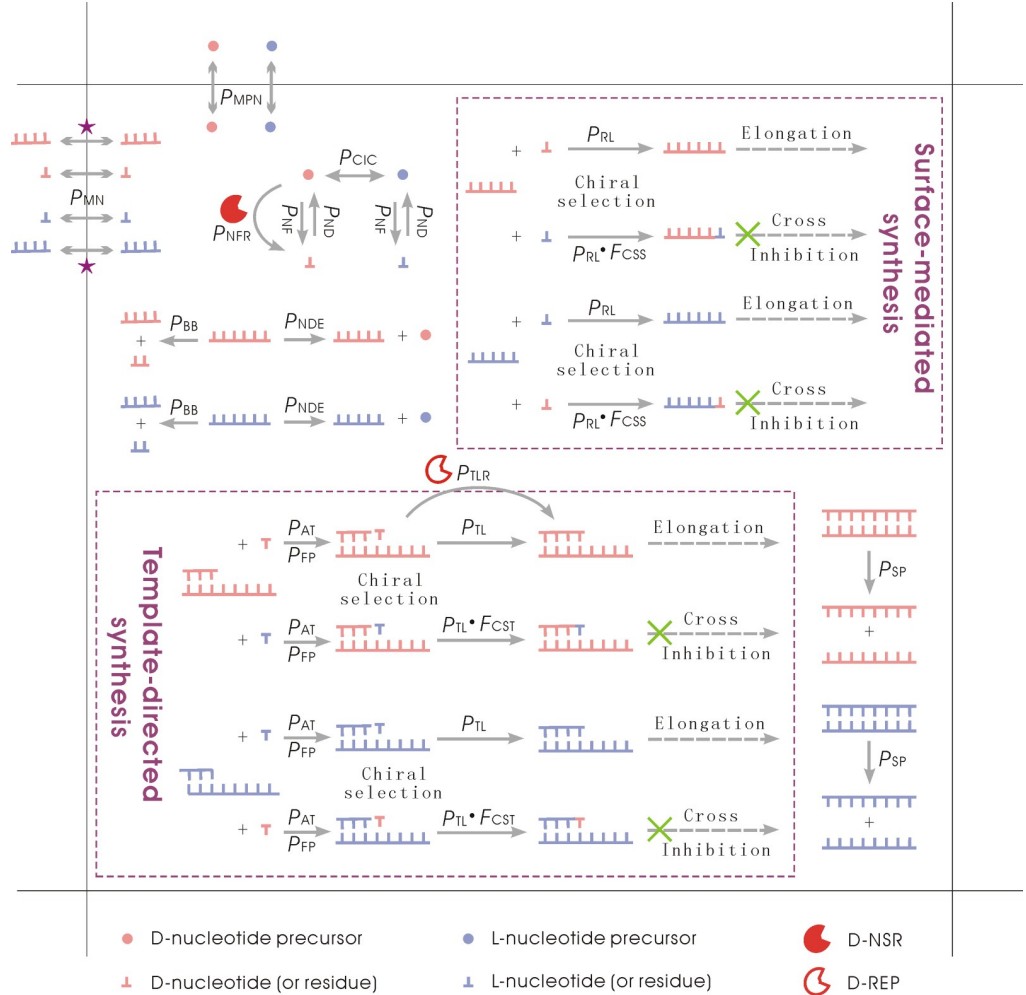

**Fig 6. Events occurring in the modeling system and relevant parameters.** The background is a grid room. Note: besides nucleotides and nucleotide precursors, RNA molecules may also move into the room or outwards (see the two stars), with a probability in positive relation to $P_{MN}$ but in reverse relation to its chain length (see Methods for details). Theoretically, both D- and L-types of NSR (nucleotide synthetase ribozyme) or those of REP (RNA replicase ribozyme) may occur in the system, and here we only show the D-type ones just for conciseness–corresponding to the cases shown in the results of this paper (Fig 4). Also for simplification, we draw here the surface-mediated synthesis and the template-directed synthesis in a way as if only monomers are able to incorporate, but actually, oligomers may also act as substrates–see Methods for a detailed description of relevant events in the model.

with the folding of the catalytic domain). When considering the nucleotide synthesis, each nucleotide precursor in the room is examined. For a precursor, if there is NSR of the same handedness within the same room, the precursor may transform into a nucleotide (certainly, also of the same handedness) with $P_{NFR}$, and simultaneously the number of such NSR available for catalysis in this room (for the present time step) is reduced by one; if there is not NSR of the same chirality type in the room, the precursor may transform into a nucleotide (of the same handedness) with $P_{NF}$ (non-enzymatic reaction).

Similarly, for the functioning of REP, each single-chain RNA molecule is checked for the ribozyme's characteristic domain (the subsequence must be in uniform handedness). The molecule is deemed to be an active REP molecule if it contains the domain but does not exceed double length of the domain. When considering the template-directed synthesis, two substrates aligned adjacently on the template may be ligated to each other under the catalysis of a

REP molecule of the same handedness (that is, the template and the two substrates and the REP must be all of the same chirality-type) with a rate of $P_{TLR}$, and simultaneously, the number of such REP available in the room is reduced by one. If there is not available REP of the required chirality-type in the room, the ligation may perform in the non-enzymatic way, with a rate of $P_{TL}$.

(**Note:** Source codes of the simulation program in C language can be obtained from Github–see Data Availability Statement. Besides the role of evidencing the reproducibility of the present study, the source codes present more details about the implementation of the model and may help readers to understand the simulation better.)

## Supporting information

**S1 Fig. The influence of the template-directed synthesis' efficiency and RNA's degradation on the chirality-deviation.** The cases are based upon the solid-line case shown in Fig 2A, i.e., the black-line case in each subfigure is actually identical to the solid-line case in Fig 2A. $P_{AT}$ (probability of an RNA template attracting a substrate) and $P_{TL}$ (probability of the template-directed ligation) are associated with the template-directed synthesis' efficiency; $P_{NDE}$ (probability of a nucleotide residue decaying at RNA's chain end) and $P_{BB}$ (probability of a phospho-diester-bond breaking within an RNA chain) are related to RNA's degradation.
(TIF)

**S2 Fig. The influence of the template-directed synthesis' chiral-selection in inducing the chirality-deviation.** A lower $F_{CST}$ means a higher rate of chiral-selection in the template-directed synthesis. The cases for $F_{CST}$ = 0, 0.5 and 1 have also been shown in Fig 2A, as the solid line, the dashed line and the dash-dotted line there, respectively.
(TIF)

**S3 Fig. The influence of the surface-mediated synthesis' chiral-selection in inducing the chirality-deviation.** The ee is shown in absolute value, that is, when L-RNA prevails (which type would prevail is actually by chance) and ee is negative, the opposite value is taken–in order to compare all the results in a clear way. A lower $F_{CSS}$ means a higher rate of chiral-selection in the surface-mediated synthesis. The blue-line case in this figure is just the solid-line case in Fig 3. That is, the cases shown here are based on a case in which $R_{n-mer} = R_{3-mer} = 20 \times R_{2-mer} = 200 \times R_{1-mer}$–note that when considering relevant experiment results, the primer effect could not be so strong [29,30], however, the high level of chirality-deviation in this case provides an ideal base upon which the influence of $F_{CSS}$ can be shown clearly.
(TIF)

**S1 Movie. A movie shown the *de novo* emergence of NSR in a background with chirality-deviation established by the surface-mediated synthesis and the template-directed synthesis.** See Fig 4A for the evolutionary dynamics of the case. The span shown here is from step $4.2 \times 10^6$ to $4.7 \times 10^6$. Also refer to Fig 5 for some snapshots of the same case, which has a broader timescale.
(MP4)

## Author Contributions

**Conceptualization:** Wentao Ma.

**Data curation:** Wentao Ma.

**Funding acquisition:** Wentao Ma.

**Investigation:** Yong Chen, Wentao Ma.

**Methodology:** Wentao Ma.

**Supervision:** Wentao Ma.

**Validation:** Wentao Ma.

**Writing – original draft:** Yong Chen, Wentao Ma.

**Writing – review & editing:** Wentao Ma.

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
