## [Decision Letter · Decision Letter 0]

10 Oct 2019

Dear Dr Ma,

Thank you very much for submitting your manuscript 'The origin of biological homochirality along with the origin of life' for review by PLOS Computational Biology. Your manuscript has been fully evaluated by the PLOS Computational Biology editorial team and in this case also by independent peer reviewers. The reviewers appreciated the attention to an important problem, but raised some substantial concerns about the manuscript as it currently stands. While your manuscript cannot be accepted in its present form, we are willing to consider a revised version in which the issues raised by the reviewers have been adequately addressed. We cannot, of course, promise publication at that time.

What is very important in the revision is that you follow the instructions from reviewer #2 and go through earlier literature more carefully and highlights what is original about this work.

Sincerely,

Alexandre V. Morozov, Ph.D.

Associate Editor

PLOS Computational Biology

Arne Elofsson

Deputy Editor

PLOS Computational Biology

[LINK]

Reviewer's Responses to Questions

**Comments to the Authors:**

Reviewer #1: This paper is an interesting simulation study of the origin of homochirality in the context of the RNA world. I have several comments concerning the reaction scheme in Figure 6.

1. The pink and blue colours of the D and L nucleotides appear as identical shades of grey when the figure is printed in black and white. A dark and light colour would be better. The fact that both D and L monomers are drawn as an L shape is also confusing, particularly when looking at a black and white print.

2. The term surface-mediated synthesis is used to describe random polymerization without a template. There does not seem to be any surface in the model, however. The environment where the random polymerization happens does not have to be a surface, as far as the model is concerned. It could be in solution, or in a lipid membrane system, etc. Therefore it might be clearer to simply call this random polymerization, or non-templated polymerization.

3. It takes some time to find the meaning of FCSS and FCST in the text. These are multipliers of the rate of addition of the opposite monomer. This could be shown by writing PRL.FCSS on the lower arrow of each pair, rather than by putting the FCSS in brackets. It is not clear that the two reaction rates are different because the same rate PRL is on both arrows.

4. It seems natural that chiral selection occurs in the templated case, but I do not understand why the surface mediated synthesis should have chiral selection. Is there any evidence from experiment that there is either slower addition of the wrong monomer or cross inhibition when polymerization is occurring on a surface? There can actually be some chiral crystals of minerals, but I am not sure if this is what is implied in the model. If we simply interpret the 'surface-mediated' reactions as general non-templated reactions, then I don't see a reason for chiral selection.

5. It seems from the diagram that precursors and nucleotides can move across the grid but not polymers. Later in the text it says that polymers can also move according to the Zimm law. It took me a long time to find this. Similarly the diagram appears to say that polymerization is one monomer at a time whereas breakdown can occur in the middle. However, eventually we can find in the text that oligomers of various sizes can also join together. These things are not very clear.

The following points are related to my own papers, therefore I will sign my name to the review.

6. The results in Fig 4 show that either the REP or NSR can appear after chirality is established by chiral selection in chemical synthesis. I agree that this makes sense, but I think we should call this chirality arising before life, not chirality arising at the same time as life. I would call the stage involving non-enzymatic templating 'chemical evolution' (as in Tupper et al [17], and Higgs (2017) J Mol Evol 84: 225-235). In the latter paper, I made a point of arguing that chemical evolution is not quite life. Life requires 'biological evolution', which would mean a specific ribozyme sequence like REP or NSR. I would call both the templated and surface mediated reactions chemical not biological. In this case, chirality arises via chemistry, and it is the presence of the chiral bias that allows the specific sequence of the chiral ribozyme to form.

Are there any other simulation results in which there is no chiral selection at the chemical stage, but a chiral ribozyme occurs which initiates the chiral bias? This would correspond to chirality arising at the same time as life, in the sense discussed by Wu, Walker and Higgs [8]. I presume that the chiral ribozyme sequence could also form in principle if the polymer synthesis were not chirally biased, but the likelihood of this would be much lower. If there are 4 monomers of one chirality, a ribozyme of length L has a relative frequency of 1 in 4^L, whereas if there are 4 monomers of both kinds, the ribozyme has a relative frequency of 1 in 8^L. The longer the ribozyme, the less likely it is to form before chirality is established. I would suppose that if the ribozyme sequence were much longer than the case of 6 which is shown, then the waiting time between the establishment of the chemical chiral bias and the origination of the ribozyme would be much longer, so it would be clear that life arose after chirality, and it would also be clear that it would be almost impossible for the ribozyme to arise if chirality did not come first.

7. A final point is that the spatial lattice in this model makes the fluctuations between L and D quite large locally, and this makes it easier to make the instability go one way or the other. This is quite important. It would be worth citing other papers in which chirality arises locally and then spreads across the lattice - e.g. Gleiser & Walker (2009) OLEB 39:479-493; Gleiser & Walker (2008) OLEB 38:293-315.

Reviewer #2: This is a technically sound paper with a limited degree of originality. Consonant with this, the coverage of previous work is rather weak. The whole paper must be reconsidered and rediscussed in the light of missing references.

First, I suggest that the Authors put a query to Google Scholar cobnerbing "origin homochiralty polymerization model" -- it is easy to find relevant papers, not cited n the present manusxript. I especially advise the authors to look at papers by Brandenburg.

Even more basic, I suggest taking a close look at this classic:

Pyranosyl-RNA: chiroselective self-assembly of base sequences by ligative oligomerization of tetranucleotide-2’,3’-

cyclophosphates (with a commentary concerning the origin of biomolecular homochirality)

from the Eschenoser group. The Authors should carefully reanalyze their results in the light of this paper, and discuss in detail the supposed novelty of their approach.

If it tuns out that the meassge is new only by having an RNA-based model in some detail, but not otherwise, the paper does not reach the publication threshold for Plos Comp Biol.

**Have all data underlying the figures and results presented in the manuscript been provided?**

Reviewer #1: Yes

Reviewer #2: Yes

PLOS authors have the option to publish the peer review history of their article (what does this mean?). If published, this will include your full peer review and any attached files.

Reviewer #1: No

Reviewer #2: No

---

## [Decision Letter · Decision Letter 1]

9 Dec 2019

Dear Dr Ma,

We are pleased to inform you that your manuscript 'The origin of biological homochirality along with the origin of life' has been provisionally accepted for publication in PLOS Computational Biology.

In the meantime, please log into Editorial Manager at https://www.editorialmanager.com/pcompbiol/, click the "Update My Information" link at the top of the page, and update your user information to ensure an efficient production and billing process.

One of the goals of PLOS is to make science accessible to educators and the public. PLOS staff issue occasional press releases and make early versions of PLOS Computational Biology articles available to science writers and journalists. PLOS staff also collaborate with Communication and Public Information Offices and would be happy to work with the relevant people at your institution or funding agency. If your institution or funding agency is interested in promoting your findings, please ask them to coordinate their releases with PLOS (contact ploscompbiol@plos.org).

Thank you again for supporting Open Access publishing. We look forward to publishing your paper in PLOS Computational Biology.

Sincerely,

Alexandre V. Morozov, Ph.D.

Associate Editor

PLOS Computational Biology

Arne Elofsson

Deputy Editor

PLOS Computational Biology

Reviewer's Responses to Questions

**Comments to the Authors:**

Reviewer #1: The previous comments have been answered carefully and I have no further comments.

Reviewer #2: The paper has been duly revised and is now much more convincing than before.

**Have all data underlying the figures and results presented in the manuscript been provided?**

Reviewer #1: None

Reviewer #2: Yes

PLOS authors have the option to publish the peer review history of their article (what does this mean?). If published, this will include your full peer review and any attached files.

Reviewer #1: No

Reviewer #2: No

---

## [Editor Report · Acceptance letter]

30 Dec 2019

PCOMPBIOL-D-19-01401R1 

The origin of biological homochirality along with the origin of life

Dear Dr Ma,

I am pleased to inform you that your manuscript has been formally accepted for publication in PLOS Computational Biology. Your manuscript is now with our production department and you will be notified of the publication date in due course.

With kind regards,

Sarah Hammond
